# Nanoscale Material Heterogeneity of Glowworm Capture Threads Revealed by AFM

**DOI:** 10.3390/molecules26123500

**Published:** 2021-06-08

**Authors:** Dakota Piorkowski, Bo-Ching He, Sean J. Blamires, I-Min Tso, Deborah M. Kane

**Affiliations:** 1Department of Life Science, Tunghai University, Taichung 40704, Taiwan; 2Center for Measurement Standards, Industrial Technology Research Institute, Hsinchu 30011, Taiwan; hopc@itri.org.tw; 3Evolution and Ecology Research Centre, University of New South Wales, Sydney, NSW 2052, Australia; sean.blamires@unsw.edu.au; 4Center for Tropical Ecology and Biodiversity, Tunghai University, Taichung 40704, Taiwan; 5Department of Physics and Astronomy, Macquarie University, Sydney, NSW 2109, Australia

**Keywords:** biological material, height image, *Arachnocampa*, biofiber

## Abstract

Adhesive materials used by many arthropods for biological functions incorporate sticky substances and a supporting material that operate synergistically by exploiting substrate attachment and energy dissipation. While there has been much focus on the composition and properties of the sticky glues of these bio-composites, less attention has been given to the materials that support them. In particular, as these materials are primarily responsible for dissipation during adhesive pull-off, little is known of the structures that give rise to functionality, especially at the nano-scale. In this study we used tapping mode atomic force microscopy (TM-AFM) to analyze unstretched and stretched glowworm (*Arachnocampa tasmaniensis*) capture threads and revealed nano-scale features corresponding to variation in surface structure and elastic modulus near the surface of the silk. Phase images demonstrated a high resolution of viscoelastic variation and revealed mostly globular and elongated features in the material. Increased vertical orientation of 11–15 nm wide fibrillar features was observed in stretched threads. Fast Fourier transform analysis of phase images confirmed these results. Relative viscoelastic properties were also highly variable at inter- and intra-individual levels. Results of this study demonstrate the practical usefulness of TM-AFM, especially phase angle imaging, in investigating the nano-scale structures that give rise to macro-scale function of soft and highly heterogeneous materials of both natural and synthetic origins.

## 1. Introduction

Adhesive materials perform important roles in locomotion, reproduction, predator defense, and prey capture in many arthropods [1,2,3,4]. Some of these materials, such as spider and glowworm capture threads [5,6,7,8,9,10], caddisfly silks [11], and hagfish slime [12,13], demonstrate impressive mechanical and adhesive properties in wet environments, either underwater or very high humidity (also see [14]). There are immeasurable applications for any synthetic materials that can mimic these properties, including in situ tissue adhesion, underwater functional glues, and weather resistant cements and adhesives [3]. A closer examination of these materials is thus warranted to assist engineers and designers in creating novel high-performance synthetic materials for a wide range of applications [3,15].

Arthropod adhesives are often complex composites that integrate both a sticky substance, e.g., aqueous glue, and solid supporting material, e.g., silk fiber [7,8,9,13]. There are many studies that describe the mechanisms, properties, and variability of their sticky glues [6,7,16,17], with fewer studies investigating the underlying materials that support them (although see [8,18]). In many cases, adhesion generated by these bio-composites is not solely defined as strong interfacial bonding but as a synergistic system involving substrate attachment and energy dissipation [8,18,19,20]. For instance, the gluey capture spiral threads of spider orb webs generate adhesion through what is known as the “suspension bridge mechanism”, where forces generated along the interface are effectively summed and transferred to the compliant and extensible core silk fiber [7,18]. Through this mechanism, the core fibers, known as flagelliform silk, perform up to 50% of the overall work of adhesion [7,18]. However, while the functional outcomes of these synergistic composites have been studied, less is known of the structural attributes, particularly at the nano-scale, that drive these properties (but see [21]).

Capture threads produced by larval fungus gnat glowworms, *Arachnocampa* spp. (Diptera: Keroplatidae) (Figure 1) [22], present a unique model for understanding structure–function relationships in bio-composite adhesives. These threads superficially display structural and functional similarities to those produced by orb web spiders. For instance, both spider and *Arachnocampa* capture threads are comprised of a pair of compliant silk fibers coated in a chemically diverse aqueous glue, with the glue secretion forming droplets to exhibit a beads-on-a-string visual appearance [7,9,10,17,23,24]. Unlike spiders, glowworms produce their sticky silk threads almost exclusively in the wet caves and temperate rainforests of New Zealand and Australia, which have low airflow and high humidity [25]. Glowworms construct a silken “curtain” of vertically hanging capture threads that are suspended from a tubular retreat along a cave ceiling or other suitable substrate [24]. Flying insects are captured in these threads as they are lured into the curtain by the blue bioluminescent glow that gives this animal its moniker [26]. Once prey is arrested the capture threads must retain the struggling insect as the larvae laboriously hauls it up for consumption [25]. Therefore, there exists a need for both strong adhesiveness and deformability of the capture threads.

Glowworm capture threads demonstrate high extensibility (strain: 47–60% strain) and compliance (Young’s modulus: 0.1 GPa) at high humidity (>90% relative humidity (RH)) while becoming stiffer (strain: 0.02–40%; Young’s modulus: 18 GPa) at lower humidity levels (30–60% RH) [9,10]. The underlying protein-based silk fibers of these threads are composed of largely hydrophilic amino acid residues, such as lysine, proline, and thymine [27], unlike silks of other arthropods that are dominated by small amino acids such as glycine, alanine, and serine [28]. Additionally, amino acid chains are reported to be assembled into cross-β-sheet crystals (aligned perpendicular to the fiber axis) [27], as opposed to the anti-parallel-β-sheet crystals of spiders and silkworms [29], at a proposed low crystallinity [27]. When exposed to water and extended, these cross-β-sheet crystals have been hypothesized to irreversibly unravel [27], as has been observed in the silk stalks of lacewings [30]. This would provide a mechanism explaining the greater mechanical compliance and extensibility of silk fibers stretched in wet conditions compared to dry conditions; however, direct evidence is lacking. To date, few empirical studies have been conducted on changes in nano-scale structures of the supporting silk fibers of glowworm threads that occur during extension.

To observe changes in nano-features of silk fibers of glowworm capture threads, we performed atomic force microscopy using stretched and unstretched threads of the Tasmanian endemic species *Arachnocampa tasmaniensis*. By using tapping mode atomic force microscopy (TM-AFM), a method long known to give information about the viscoelastic material properties of polymers [31], we were able to generate both height and phase angle spatial maps of regions of the surface of capture threads. For polymers, it is reported that the phase angle change is dominated by tip-sample viscoelastic deformation [32] and therefore is a good indicator of variations in the viscoelastic properties of a material. This method has been used extensively for the study of heterogeneous polymer samples [32]. However, what has been largely overlooked in experimental application, though well established and demonstrated in principle [33,34], is the substantially higher spatial resolution of nano-scale features, associated with varying viscoelastic properties, in the phase angle map of an area of a sample as compared with what is visualized in its height map. While the height map does an adequate job of visualizing surface substructure and local geometry, phase imaging can provide visualization of variation in material density and viscoelastic properties that may or may not correlate with surface structure. For instance, a rough surface composed of a homogenous material shows low variation in phase contrast and lower resolution than what can be observed in the height map, as demonstrated in PDMS [35] and glass [36,37]. However, the slope of a surface is still a contributing factor in determining phase shift [31]. Hence, we used TM-AFM here to assess changes in nano-scale features before and after stretching of glowworm capture threads by determining differences in relative, qualitative viscoelastic properties. 

## 2. Results

### 2.1. Tapping Mode AFM and QNM Modulus Mapping of Glowworm Threads

TM-AFM of *A. tasmaniensis* glowworm silk threads revealed nano-scale surface features displaying a diversity of size, structure, and orientation (Figure 2). In general, thread samples demonstrated predominately globular features (Figure 2A,B), a mix of globular and elongated features (Figure 2C,D), or predominantly vertically elongated features (Figure 2E,F). Height images demonstrate that silk fibers are generally curved, although relatively flat across the lateral axis (10–60 nm height range) (Figure 2A,C,E). Compared with height images, phase imaging provided greater spatial resolution of nano-scale features (Figure 2). Quantitative nanomechanical (QNM) modulus mapping of two regions of an unstretched fiber section indicated that lower phase angles corresponded to greater relative modulus (Figure 3). Average values determined from 25 indentation points in each region showed that modulus values were ~1.60 times greater in the darker shaded (lower phase angle) regions (Figure 3B). This is further supported by the QNM modulus mapping, which shows qualitative agreement in the difference in modulus between regions (Figure 3C). However, the 10 points of the QNM results in region 2 that are closest to the right hand edge, where the adhesion is higher (Figure 3D), as per the top left region, have a smaller average Young’s modulus of 2.62 MPa. The ratio of this subset of region 2 to region 1 is then reduced to ~1.37. The uncertainty is estimated conservatively as 0.2, allowing the possibility that adhesion effects may not have been fully taken into account leading to the estimated possible range of the ratio of 1.4 ± 0.2.

### 2.2. Fast Fourier Transform Analysis of Phase Angle Images of Glowworm Threads

Variation in feature size, structure, and orientation was identified between stretched and unstretched glowworm fibers, as well as within individuals and even fibers (Figure 4). FFT analysis of phase images reflected the observed variation in feature characteristics, and qualitative differences between unstretched and stretched threads were observed (Figure 5). Unstretched samples revealed largely globular features with varying spatial frequencies (>5 nm/cycle) (Figure 5A,B) or unaligned, elongated features with similar spatial frequencies (10–15 nm/cycle) (Figure 5C,D). In stretched threads, vertically aligned features with similar spatial frequencies (10–15 nm/cycle) (Figure 5E,F) were predominately observed. Analysis of FFT of phase images of stretched (N individuals: 10; n threads: 40) and unstretched (N individuals: 9; n threads: 40) threads revealed several similarities in feature characteristics (Figure 6, Appendix A). The far diffuse cloud (Figure 6A) was defined as the apparent shift in grayscale color gradient that was observed at 4–5 nm/cycle (arrow, inset) and roughly resembling a cloud in one of four configurations. The near dense cloud (Figure 6B) was defined as the apparent shift in grayscale color gradient that was observed at 15–20 nm/cycle (arrow, inset) and roughly resembling a cloud in one of four configurations. Arcs (Figure 6C) varied in angular length and number but were obvious features that were observed at 10–20 nm/cycle (arrows, inset). All FFT images are shown as generated (including in the full set of images Appendix A). Image processing of the FFT, such as contrast enhancement, can be used to more clearly differentiate the most prevalent spatial period features in the images, though not used herein. Further information on interpreting FFT images is available in Appendix A. Short arcs were defined as an angular distance of less than 90° and long arcs as greater than 90°. Arc orientation (Figure 6D) varied around the full 360° range (arrows, inset). Vertical orientation was defined as arcs fully crossing the horizontal axis at 0° or 180°. Note an arc on the horizontal axis of the FFT is generated by vertically oriented periodic spatial structure of a spatial period that can be measured from the FFT, and it is also observable in the original phase image. Horizontal orientation was, thus, arcs crossing the vertical axis at 90° or 270°. Arcs in the northeast–southwest (NE-SW) and northwest–southeast (NW–SE) orientation neither crossed the vertical nor the horizontal axis. For example, the inset example was characterized as NE–SW (Figure 6D). The presence of paired arcs was 66% greater (Figure 6C), and the vertical orientation of arcs was 100% greater (Figure 6D) in stretched threads compared with unstretched threads. These results support the general observation of greater vertical orientation of features in the phase images for stretched threads (See Figure 4).

## 3. Discussion

TM-AFM phase angle images of samples of dried *A. tasmaniensis* glowworm silk on glass slides revealed nano-scale variations in the geometry and viscoelastic properties of the material at and near the surface (Figure 2 and Figure 4). FFT analysis of these images has allowed prevalent length scales, forms, and orientation of the forms to be determined (Figure 5 and Figure 6). Globular and fibrillar variations with widths of 10–15 nm are the most prevalent feature observed (Figure 2 and Figure 4). This is against a background that shows features of all size scales down to ~5 nm/cycle in elastic modulus variations (Figure 5) with 2 nm/cycle as the lower limit of features that could be shown by the FFT process. Within the range of features observed, we demonstrated qualitative differences between unstretched and stretched threads and emphasize that quantitative characterization was not possible within the scope of this study. 

The physics of the surface-tip interaction in TM-AFM can be complex, as it reflects energy dissipation, which incorporates adhesive, frictional, and viscoelastic properties of a sample surface and air damping of the oscillation of the cantilever [38]. Further complications can arise due to surface contaminants such as moisture and hydrocarbons, which may modify the surface [39]. All these possibilities need to be taken into account to make quantitative measurements. It is, thus, very challenging to achieve accurate elastic modulus values resolved on the few nanometer-length scales as observed for glowworm silk threads. As such, most studies using AFM to investigate properties of spider or insect silks have primarily relied on height mapping [40,41,42,43] rather than phase-angle images (but see [44,45]). However, height mapping is only capable of informing on variation in the geometric attributes of the surface of a material. Phase mapping, even with a non-quantitative approach as in this study, is capable of indicating variations in density and/or composition along a flat but heterogenous material (see [35,36,37]). 

In this study, phase mapping displayed greater spatial resolution than height mapping of features observable in both TM-AFM images and revealed features indistinguishable in height images (Figure 2). This indicates that variation in surface geometry may have contributed to the generation of the observed phase contrast [31] and/or may be correlated with variation in viscoelastic properties. Regardless, these nano-scale features, e.g., globular and elongated fibular patterns (Figure 2), could represent substructures within the silk fiber of these capture threads, as have been identified in spider and silkworm silks [29,40]. However, the relationship between surface structure and viscoelastic properties due to underlying variation in surface roughness, material density, composition or molecular arrangement is complex and requires further study to resolve. Additionally, the values of modulus obtained from QNM and its spatial variation are different from the average quantitative values measured by standard tensile testing (18.38 GPa at 30% RH, 0.1 GPa at >90% RH) [9], which makes further contextualization and interpretation of our results difficult. This disagreement is not unexpected and could be due to a number of factors, such as greater axial versus lateral modulus in silk fibers [46,47] and differing material chemistry in different regions [48,49], which highlights the need for further investigation using more quantitative methods to make more conclusive assertions. 

An open question that remains is of what significance do the nano-scale features have in the biological function of *Arachnocampa* glowworm capture threads? Our findings support previous reports of low crystallinity [27], as threads appear to exhibit a higher amount of relatively soft features covering a larger degree of area across a given region of thread. A softer, more elastic component could facilitate prey capture by decreasing oscillations resulting from insect impact and movement by storing more energy relative to external dissipation from viscous air drag. A partial re-orientation of some fraction of the fibrillar features to a more vertical orientation during stretching could also assist with reversible fiber lengthening and high mechanical compliance observed in these threads [9]. Frictional forces may be applied between nano-scale features during possible stretch-induced re-orientation, which would allow for internal dissipation during deformation. Movement of nano-features may be facilitated by absorbed water, which has been shown to increase molecular mobility in silks of other animals [50,51]. However, further work needs to be conducted to make more direct links between material structure and functional performance of these glowworm threads, as has been done in spider silks [52,53,54,55]. When studying the adhesive performance of arthropod silks or other biological materials though, it is imperative to consider the added influence of energy dissipation that may occur in the system [8,18,19,20]. 

A significant finding of this research was the demonstrated practical application of TM-AFM phase angle imaging to interrogate variations in the material properties of a biological material at length scales of a few nanometers. Despite an inability to quantify the modulus variations observed, which could be a future task for those who are expert in the physics of TM-AFM, we were able to discern qualitative differences between relative modulus variations in stretched and unstretched glowworm threads. The increased vertical orientation of fibrillar nano-features seems to be correlated with the macroscopic stretching of these threads, which may allow increased energy dissipation during prey capture. TM-AFM, phase imaging in particular, should see increased use as a powerful tool for investigating heterogeneous material properties at the nanoscale in biological materials such as silks. Ultimately, understanding the structure–function relationships involved in natural materials using tools such as TM-AFM can inform the design of next-generation synthetic adhesives and materials.

## 4. Materials and Methods

### 4.1. Capture Thread Collection

A total of 20 samples of capture threads produced by *Arachnocampa*
*tasmaniensis* were collected from 10 individuals from the entrance and twilight zones of the Bradley Chesterman caves in Southwest National Park, Tasmania, Australia, in October 2017. The natural adhesiveness of vertically hanging glowworm capture threads allowed us to collect pristine capture thread fragments across the short length of glass microscope slides (75 × 26 mm). Once thread fragments were affixed to the slide, the upper section of the thread was cut away from the glowworm snare with scissors. Two samples were collected per individual along each side of the glass slide—one in its native, unstretched state and one stretched prior to collection. For stretched samples, the lower end of a thread was first adhered to the tip of a finger and extended at a strain rate of between 10% and 20% s^−1^ up to 120–125% of original length without fracturing the thread. This value was chosen as it is just below the average breaking strain (128%) of extended threads in natural cave environments [9]. Samples were stored in airtight containers and transported to Tunghai University, Taichung, Taiwan, within two weeks of collection where samples were stored at ambient laboratory conditions (25 °C, 45% RH).

### 4.2. Atomic Force Microscopy

Atomic force microscopy (AFM; Dimension Icon Bruker, Billerica, MA, USA) was conducted on glowworm thread samples from December 2017 to March 2018 at the Center for Measurement Standard at the Industrial Technology Research Institute, Hsinchu, Taiwan. Height and phase images of the silk threads were measured simultaneously through the AFM dynamic standard tapping mode. Regions along the silk fiber axis between glue droplets (Appendix A) were chosen for imaging, as these were areas of silk fiber least likely to be fully coated in glue. SSS-NCHR SuperSharp cantilevers were used with a nominal cantilever spring constant (k) of 42 N/m and resonance frequency of 330 kHz. The typical tip radius was 2 nm. The amplitude setpoint was automatically determined by the AFM device and fell in the range of 350–375 mV. The scan rate was 1 Hz, and the resolution was 512 × 256 for all AFM images. Compared with the height image, the sample phase image depicted additional surface details and was selected to illustrate the changes within the silk thread surfaces in AFM scanning. Regions of 500 × 500 nm were selected, as spatial resolution was optimized compared with 100 × 100 nm (Appendix A) or 1 × 1 μm (Appendix A). Additional height and phase images were obtained of the glass slide surface and glue droplet spread across the glass surface without silk fiber (Appendix A). A total of 98 height and phase images were generated of unstretched (N threads: 10; n images: 47) and stretched threads (N threads: 9; n images: 47), surface of glass slides (N slides: 2; n images: 2) and glue spread on glass (N threads: 2; n images: 2) (See Appendix A for full set of images).

The exact stiffness of the cantilever tip was calibrated through indentation on a sapphire substrate and then determined using the thermal tune method [56]. This was done before executing the scanning mode, as the stiffness value is required to quantitatively set the surface–tip interaction force. The detailed calibration procedure is as follows: firstly, the X and Y voltages applied to the piezo tube were held constant, and the ramped Z led to cyclically varied distance between tip and substrate, causing cantilever deflection. The complete cycle of the interaction process occurs as the cantilever approaches and retracts from the substrate (Appendix A). Deflection was calculated by determining the slope of the deflection error (mV) over deflection sensitivity (V/nm) curve during retraction (Appendix A). In addition, an accurate estimate of the cantilever stiffness involving the measurement of the cantilever’s mechanical response to thermal noise was used by thermal tune method. In this mode, twenty-five force curves within the selected image area were deduced. Sneddon’s model was then fitted to each curve, and the corresponding modulus was deduced [57]. The sample stiffness, which was obtained from the unloading curves, was considered in this model.

Modulus was qualitatively determined from a single capture thread using standard point nanoindentation (See Appendix A for test parameters) and AFM quantitative nanomechanical (QNM) mapping mode [58,59]. First, a 500 × 500 nm section of the capture thread, which was previously mapped using TM-AFM, with regions of variable phase contrast, was mapped using QNM. Second, two 5 × 5 grids were generated along known regions with contrasting phase angle ranges determined by TM-AFM, i.e., a region with high versus low relative phase angle values and modulus ranges determined by QNM. Average values derived by point indentation of these regions, along with QNM mapping, were used as the best available means to establish how relative modulus varied in relation to the observed range of phase-angle measurements. These measurements were not meant to formally calibrate the observed phase contrast, which was beyond the scope of our study, but were to interpret the correlation between phase angle and modulus on a lower to higher basis. 

### 4.3. Fast Fourier Transform Analysis

Phase images of 500 × 500 nm sections of stretched (N threads: 10; n images: 40) and unstretched threads (N threads: 9; n images: 40) were analyzed using fast Fourier transform processing in ImageJ [60]. The calibration features available in ImageJ were used to interactively record the spatial periods and angles of orientation associated with higher brightness features in the fast Fourier transform (FFT) (Appendix A). Features of the raw FFT images were broadly characterized, and presence/absence data were collected. Information collected included: (i) presence and overall shape of a diffuse cloud with discernable edge at 5–6 nm/cycle, (ii) presence and overall shape of a diffuse cloud with a discernable edge at 10–20 nm/cycle, (iii) presence and overall shape of a dense cloud with a discernable edge at >20 nm/cycle, and (iv) presence, length, and orientation of arcs at 10–15 nm/cycle. See Appendix A for further description of the significance of the features observed in the FFT of the phase angle images.

## Figures and Tables

**Figure 1 molecules-26-03500-f001:**
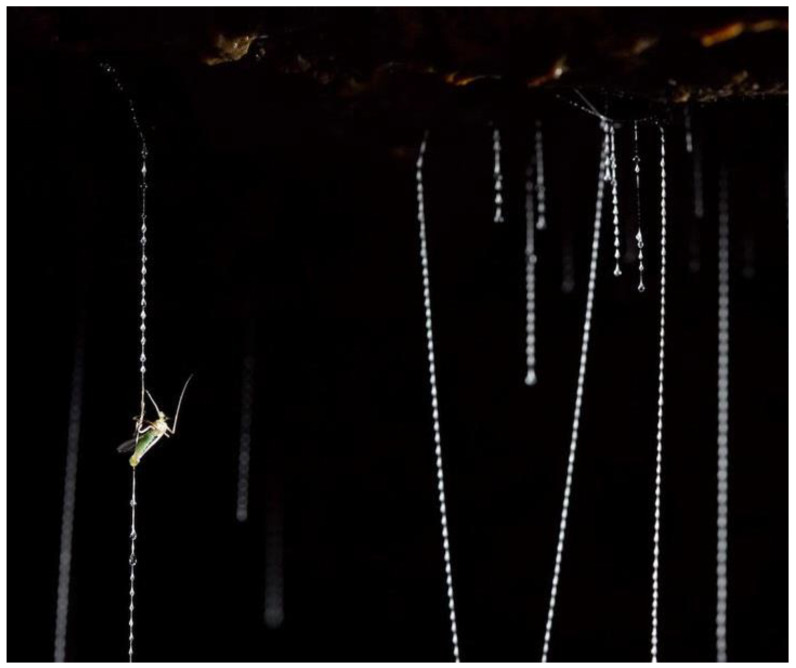
Vertically hanging *Arachnocampa tasmaniensis* silk threads with a captured insect prey. Copyright: Bookend Trust/SIXTEEN LEGS. Photo credit: Joe Shemesh.

**Figure 2 molecules-26-03500-f002:**
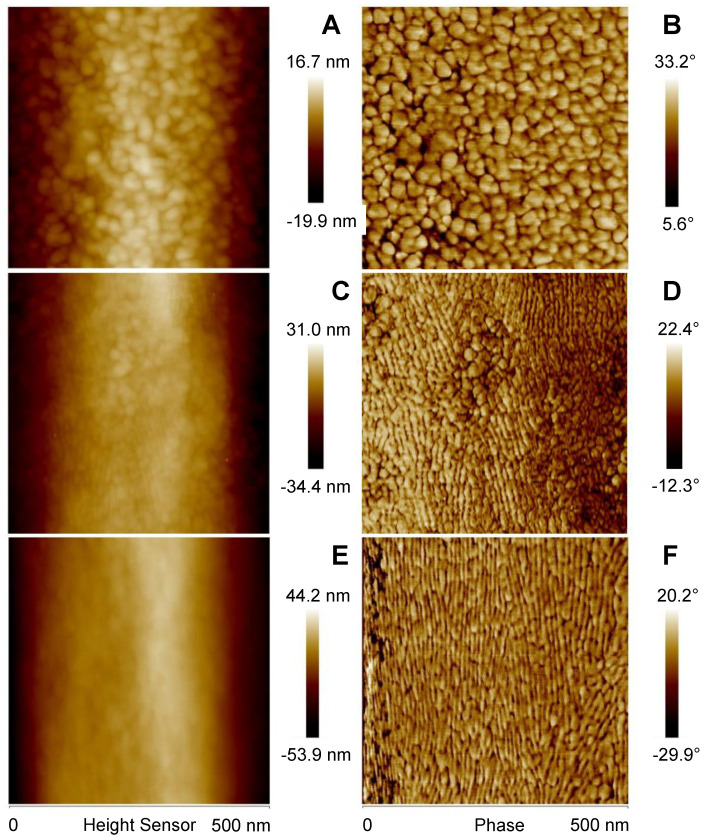
Enhanced resolution of nano–scale features of *Arachnocampa tasmaniensis* silk threads from phase imaging compared with height imaging. Simultaneous height (**A**,**C**,**E**) and phase images (**B**,**D**,**F**) were taken from three samples of glowworm silk threads.

**Figure 3 molecules-26-03500-f003:**
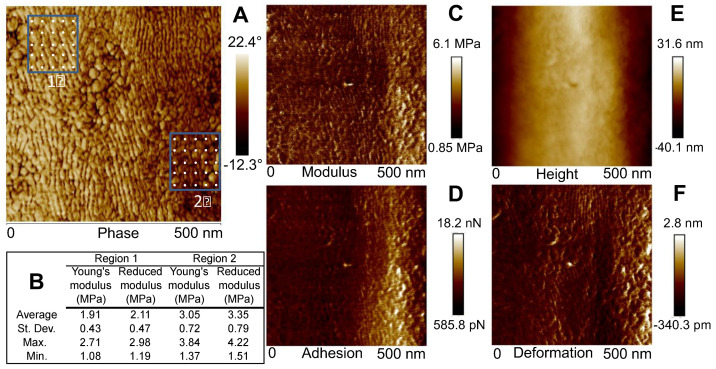
Results of quantitative nanomechanical (QNM) and tapping mode (TM) AFM imaging along a 500 × 500 nm area of an unstretched silk thread from *Arachnocampa tasmaniensis*. QNM was conducted in a 5 × 5 grid pattern in two distinct regions of the thread sample previously mapped through TM–AFM (**A**) and showed differences in modulus values (**B**). Other QNM channels (adhesion, modulus, deformation) also show differences between regions (**C**–**F**).

**Figure 4 molecules-26-03500-f004:**
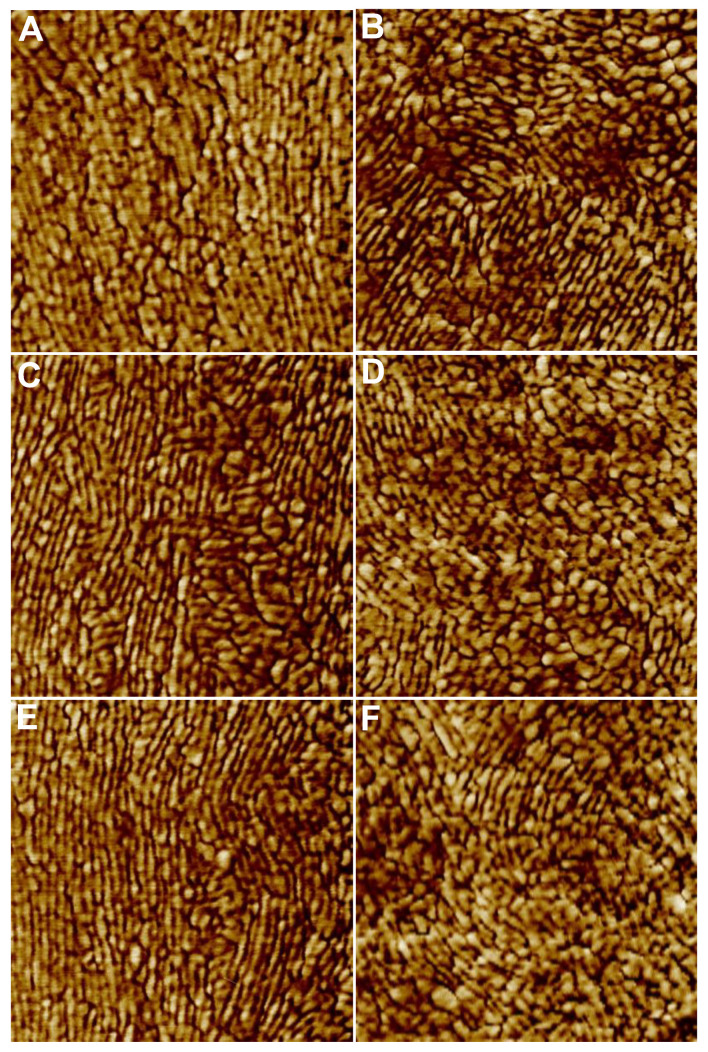
Comparison of phase images of (**A**,**C**,**E**) stretched and unstretched (**B**,**D**,**F**) *Arachnocampa tasmaniensis* glowworm threads from the same individual.

**Figure 5 molecules-26-03500-f005:**
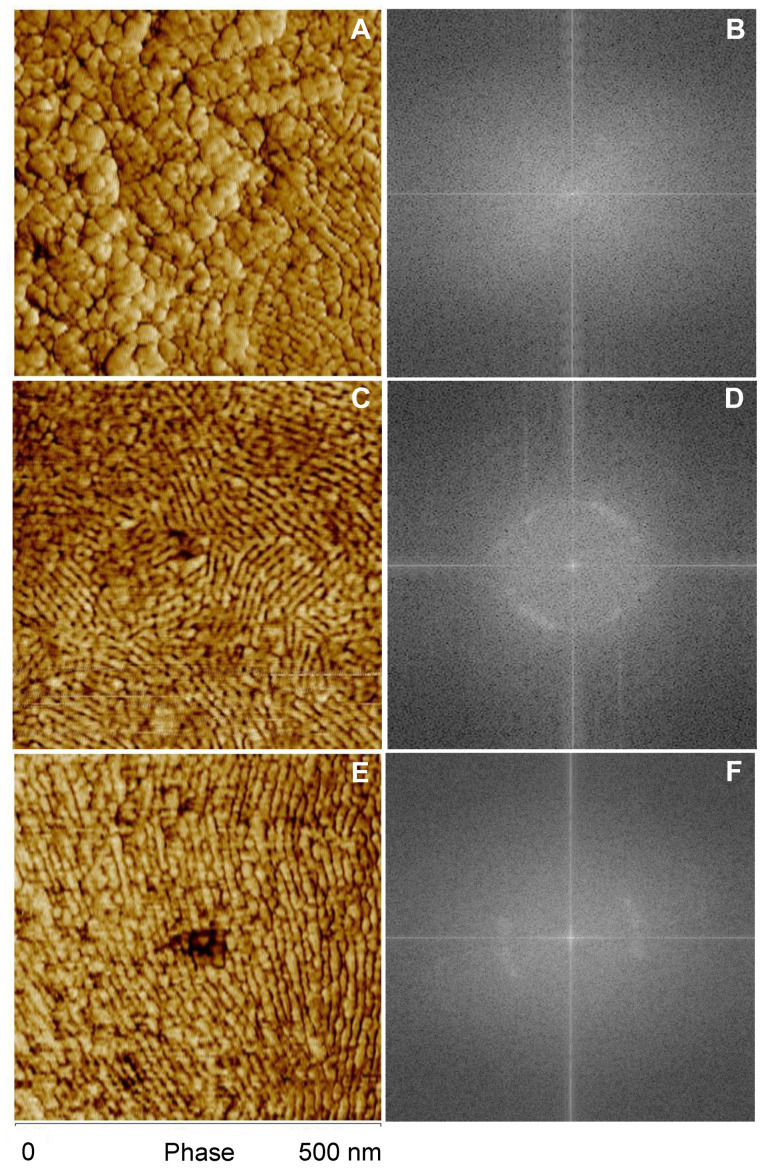
Phase images (**A**,**C**,**E**) and corresponding fast Fourier transform (FFT) results (**B**,**D**,**F**) taken from two unstretched samples (**A**–**D**) and one stretched sample (**E**,**F**) of *Arachnocampa tasmaniensis* glowworm thread.

**Figure 6 molecules-26-03500-f006:**
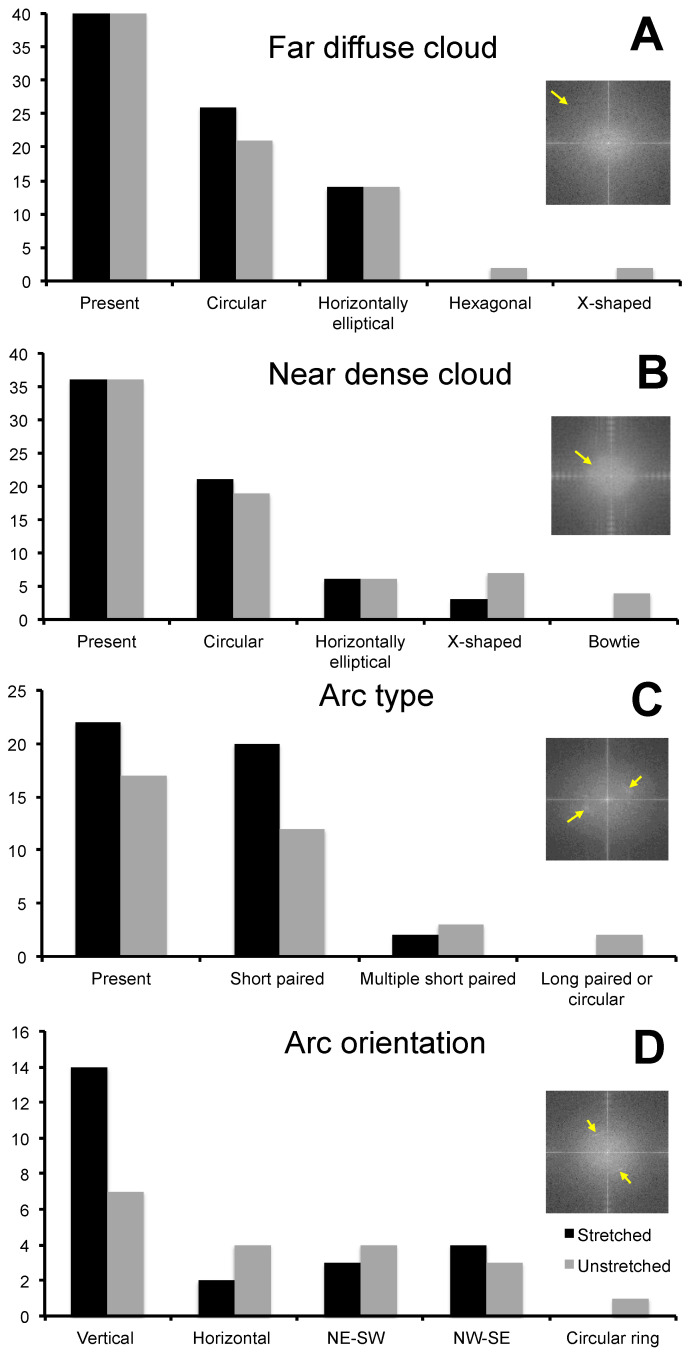
Characterization of qualitative features from fast Fourier transform analysis of phase images of stretched and unstretched silk threads from *Arachnocampa tasmaniensis* glowworms. Arrows within inset panels are used to demonstrate the edge of the far diffuse cloud (**A**) and near dense cloud (**B**), arc presence (**C**) and orientation (**D**).

## Data Availability

Data supporting reported results can be found Appendix A.

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
