# Peer review of "Nanoscale Material Heterogeneity of Glowworm Capture Threads Revealed by AFM"

_molecules, 2021, doi:10.3390/molecules26123500_

Round 1

Reviewer 1 Report

The article by D. Piorkowski et al. presents the analysis of AFM phase images via FFT as a way to distinguish features encountered in the morphology of either stretched or unstretched silk fibers.

In my opinion, the results are poorly presented. The authors observe morphological and mechanical differences but these are not discussed in context. The authors merely state that there are differences.

line 140. Comment: It would have been preferable to calibrate on the sapphire after the measurements to avoid damaging the tip.

line 150. Comment: it is not clear if the mechanical 25 points grid was performed for all fibers, or if only for the one presented. If it was performed for all fibers, perhaps a histogram is missing?

line 154. It would be informative to provide more information on the disagreement between QNM and standard mechanical testing. The authors only refer to the later values. 

The last sentence in this section is not very clear to me. Does it simply mean that the authors use QNM despite not being in agreement with standard mechanical testing, or it means that the authors use QNM to calibrate phase contrast? Or is it only to stress that there is a one-to-one relation between phase differences and mechanical properties?

Relative to figure 2: Naturally the phase images show enhanced contrast, nonetheless, the comparison of phase with topography without removing the overall topography envelope is not really fair!

In figure 3, the authors refer to TM channels, however, adhesion, modulus and deformation, are not really TM channels. These are model dependent and instrument dependent, AFMs from other brands may not have these channels and/or the calculations performed to obtain them may be different. These are calculated from the TM amplitude and TM phase (and hopefully from the deflection) which is really what is measured. In my opinion, since these signals bring no new relevant information they should not be presented. Otherwise, it must be explained how these are obtained. Additionally, the scale in these images seems very poorly adjusted.

As for the QNM, there should be a reference pointing to how these calculations are done.

line 281: a softer more elastic (elastic means energy is conserved) component would not dissipate the energy of the impact at least not directly. It would have an impact on the oscillation frequencies, and on the ratio of viscous drag (from air)  to energy stored therefore decreasing the number of oscillations before the thread stops… 

However, I expect that the fibers contain cleverer ways of dissipating energy, for instance, if the length after insect strike is increased then some features move past each other which implies friction, and would be a direct form of energy dissipation. Have the authors investigated this possibility?

The authors claim that their main result is the opportunity to use TM-AFM phase differences to interrogate variations in the material properties of the glowworm threads, however, phase contrast has been used for quite awhile now and is well known to enhance the contrast in samples that are not homogeneous. This is present in many publications and in books that address AFM. Careful quantification of these phase differences is difficult though.

In my opinion the stress must be put on the characterization of these threads by phase-contrast rather than on the very well-known fact that the phase difference enhances contrast in cases like this.

Author Response

Response to Reviewers:

We have used track changes to highlight revisions made to our manuscript.

Reviewer 1:

The article by D. Piorkowski et al. presents the analysis of AFM phase images via FFT as a way to distinguish features encountered in the morphology of either stretched or unstretched silk fibers.

Author response: We thank the reviewer for useful comments and suggestions that helped us improve the quality of our manuscript.

In my opinion, the results are poorly presented. The authors observe morphological and mechanical differences but these are not discussed in context. The authors merely state that there are differences.

Author response: We have expanded our Discussion throughout, particularly adding to paragraphs 2 and 4, as well as a new third paragraph, to provide broader context.  

line 140. Comment: It would have been preferable to calibrate on the sapphire after the measurements to avoid damaging the tip.

Author response: The execution of the stiffness calibration of the cantilever before measurement is necessary because the stiffness value is required to quantitatively set the surface-tip interaction force during both modulus mapping and 25 points grid indentations in QNM mode. We have a new second sentence to the second paragraph of section 2.2 of the Methods.

line 150. Comment: it is not clear if the mechanical 25 points grid was performed for all fibers, or if only for the one presented. If it was performed for all fibers, perhaps a histogram is missing?

Author response: QNM was not performed on all samples, only on a single thread in two different regions. We have made this clear in the first sentence of the third paragraph of section 2.2 of our Methods.

line 154. It would be informative to provide more information on the disagreement between QNM and standard mechanical testing. The authors only refer to the later values.

Author response: We removed this statement from our Methods and have moved it to the fifth sentence of the third paragraph our Discussion where we also provide further information and context.

The last sentence in this section is not very clear to me. Does it simply mean that the authors use QNM despite not being in agreement with standard mechanical testing, or it means that the authors use QNM to calibrate phase contrast? Or is it only to stress that there is a one-to-one relation between phase differences and mechanical properties?

Author response: QNM and point nanoindentation was primarily used to determine which end of the phase angle range corresponds to relatively higher vs. lower modulus values and corresponding features. It was not meant as a formal calibration of any sort. Also, we are not implying which kind of relationship (linear/non-linear) phase angle change has with mechanical properties as we do not have sufficient information. We have revised the third paragraph of section 2.2 of our Methods to address reviewer concerns.

Relative to figure 2: Naturally the phase images show enhanced contrast, nonetheless, the comparison of phase with topography without removing the overall topography envelope is not really fair!

Author response: Topographies have been presented without image processing or scale adjustment. These kinds of alterations would make our height images no longer meaningful as they would not truly be height in the literal sense, but some kind of adjusted height. We also believe it is more important to present to readers the original images, as obtained. Also, we dispute the notion that surface structure alone would lead to enhanced phase contrast. Several studies demonstrate that along a rough but homogeneous surface there is variation in height mapping, with little to no variation in the phase angle map, see refs 35,36,37. The identical material properties across a homogenous but rough surface would not register a change in lead or lag in the phase response and would therefore not show greater contrast. However, if surface structure is correlated to differences in material density or changes in modulus, then it may not be possible to disambiguate. We have made mention of this possibility in the sixth and seventh sentence of the fifth paragraph of the Introduction and sixth and seventh sentence of the second paragraph of the Discussion.

In figure 3, the authors refer to TM channels, however, adhesion, modulus and deformation, are not really TM channels. These are model dependent and instrument dependent, AFMs from other brands may not have these channels and/or the calculations performed to obtain them may be different. These are calculated from the TM amplitude and TM phase (and hopefully from the deflection) which is really what is measured. In my opinion, since these signals bring no new relevant information they should not be presented. Otherwise, it must be explained how these are obtained. Additionally, the scale in these images seems very poorly adjusted.

Author response: These images are from QNM channels, not from TM channels. We corrected this error in the text of our draft in the Methods, Results and Figure 3. Although these quantified mechanical values from QNM channels would vary with different models and instruments, the trends of these values between different regions should be consistent. Scale bar was re-designed for better clarity as suggested. See Figure 3.

As for the QNM, there should be a reference pointing to how these calculations are done.

Author response: We have included reference #41 that describes how calculations were performed.

line 281: a softer more elastic (elastic means energy is conserved) component would not dissipate the energy of the impact at least not directly. It would have an impact on the oscillation frequencies, and on the ratio of viscous drag (from air) to energy stored therefore decreasing the number of oscillations before the thread stops…

Author response: We thank the reviewer for pointing this out. We have amended the third sentence of the fourth paragraph of our Discussion to reflect reviewer comments.

However, I expect that the fibers contain cleverer ways of dissipating energy, for instance, if the length after insect strike is increased then some features move past each other which implies friction, and would be a direct form of energy dissipation. Have the authors investigated this possibility?

Author response: We also believe this could be a possibility and while we do not have direct experimental evidence to address this we have incorporated these ideas into the fifth sentence of the fourth paragraph of our Discussion.

The authors claim that their main result is the opportunity to use TM-AFM phase differences to interrogate variations in the material properties of the glowworm threads, however, phase contrast has been used for quite awhile now and is well known to enhance the contrast in samples that are not homogeneous. This is present in many publications and in books that address AFM. Careful quantification of these phase differences is difficult though.

Author response: We agree that it is known that phase angle from tapping mode AFM can be used for enhanced contrast of heterogenous surfaces of materials in theory, it has not been demonstrated extensively in practice. We have made changes to our text to reflect this in the fifth sentence of the fifth paragraph of the Introduction.

In my opinion the stress must be put on the characterization of these threads by phase-contrast rather than on the very well-known fact that the phase difference enhances contrast in cases like this.

Author response: We have provided an enhanced emphasis on the characterization of these threads while still shedding light on the fact that TM-AFM is not extensively used, practically, to demonstrate surface heterogeneity through revisions throughout the Discussion.

Reviewer 2 Report

The Authors report an interesting study of mechanical properties of glowworm silk samples, that is undoubtedly an important topic for both physics and natural science knowledge. An impressive amount of measurements has been performed on several samples and over a long time period. Morphological characterization is extensive and appears as exhausting and carried out with scientific rigour. Unfortunately, the same cannot be stated for the nanomechanical characterization, that is claimed to be the focal topic of the paper, being also mentioned in the title.

I regret that the use of tapping mode (TM) of AFM, specifically the phase information, is not so straightforwardly related to elastic moduli, adhesion or other mechanical properties of the surface. As also mentioned in the paper, TM-AFM suffers from complex physical interpretation of results, as also demonstrated in cited references (there is confusion in some reference numbering, however Ref. 31 explains the subject with enough detail). It is well known that TM phase signal is affected by local dissipation as well as from the local geometry of the sample, that could be possibly the dominant contribution in all shown images.

As an example, it is well known that the TM phase values are strongly affected by the TM amplitude setpoint, however in the paper there is no mention to how the amplitude setpoint was chosen, whether it is the same on the many scans, or contrarily, haw was it adjusted in order to get always about the same phase angle response over the many images shown both in the paper and supplementary material. As it can be evinced from the basics of TM phase imaging (Ref 31 or many others available), the value of phase angle is related to the interaction regime (attractive or repulsive), and to both conservative and dissipative components of the interaction. However, such a relation is rather involved, and conclusions cannot usually be drawn straightforwardly from the acquired data. In the absence (or with a constant value) of dissipative interaction, phase contrast is only ruled by conservative interactions, that is the reason why the phase image appears usually as a sort of “spatial derivative” of the topographic image, with enhanced contrast at the borders of the structures, that highlights the shape of objects that would be more difficult to tell from the sole topography. In this case, it is not correct to assign to the phase contrast any meaning related to mechanical properties, since the same contrast is observed as well on homogeneous (but rough) surfaces.

Quantitative nanomechanical mapping (QNM) is claimed to provide more accurate information about local mechanical properties, possibly also taking into account the contribution of local morphology. Even by assuming that the outcome of this technique could provide more significant values of local nanomechanical properties, in this work QNM is applied by just sampling on a few points of the surface, and not by full mapping with nanometer-scale resolution, therefore it is difficult to correlate the nanomechanical measurement with the related morphology (that is, to compare systematically the values on top of the structures with the ones in the regions between structures).

In conclusion, although a lot of interesting information on the sample morphology is provided, the main aim of the paper, that is, to gain insight on nanomechanical properties of the investigated samples, seems not supported by evidence. The assessment of nanomechanical properties cannot be demanded exclusively to TM phase imaging, but it should be supported by more thorough exploration, by QNM or other nanomechanical investigation tools, by showing the claimed correlations between structure and functionality.

Author Response

Response to Reviewers:

We have used track changes to highlight revisions made to our manuscript.

Reviewer 2:

The Authors report an interesting study of mechanical properties of glowworm silk samples, that is undoubtedly an important topic for both physics and natural science knowledge. An impressive amount of measurements has been performed on several samples and over a long time period. Morphological characterization is extensive and appears as exhausting and carried out with scientific rigour. Unfortunately, the same cannot be stated for the nanomechanical characterization, that is claimed to be the focal topic of the paper, being also mentioned in the title.

Author response: We have changed the title and made adjustments to our Introduction in paragraph 5, Methods in the third paragraph of section 2.2 and paragraph 3 of the Discussion to reflect suggestions made by the reviewer. Our goal was not to provide quantitative nanomechanical characterization, rather, to use nanomechanical characterization as a means to correlate the change in phase angle with heterogeneity in the material properties, to draw attention to the nanoscale morphology of these variations, and to quantify the sizes of these forms.

I regret that the use of tapping mode (TM) of AFM, specifically the phase information, is not so straightforwardly related to elastic moduli, adhesion or other mechanical properties of the surface. As also mentioned in the paper, TM-AFM suffers from complex physical interpretation of results, as also demonstrated in cited references (there is confusion in some reference numbering, however Ref. 31 explains the subject with enough detail). It is well known that TM phase signal is affected by local dissipation as well as from the local geometry of the sample that could be possibly the dominant contribution in all shown images.

Author response: We have corrected reference numbering and apologise for the confusion. Local geometry could affect the phase map of a sample if there is non-homogenous distribution of material properties along its surface. For instance, a rough surface would show variation in along its height map, but would be largely homogenous in its phase map, see refs 35,36,37. While we cannot quantify the differences we observe in the phase map nor can we state whether differences in material properties are correlated to local geometry, we can say that local geometry cannot solely be the determinant of the observed phase contrast. We have made mention of this possibility in the sixth and seventh sentence of the fifth paragraph of the Introduction and sixth and seventh sentence of the second paragraph of the Discussion.

As an example, it is well known that the TM phase values are strongly affected by the TM amplitude setpoint, however in the paper there is no mention to how the amplitude setpoint was chosen, whether it is the same on the many scans, or contrarily, how was it adjusted in order to get always about the same phase angle response over the many images shown both in the paper and supplementary material. As it can be evidenced from the basics of TM phase imaging (Ref 31 or many others available), the value of phase angle is related to the interaction regime (attractive or repulsive), and to both conservative and dissipative components of the interaction. However, such a relation is rather involved, and conclusions cannot usually be drawn straightforwardly from the acquired data. In the absence (or with a constant value) of dissipative interaction, phase contrast is only ruled by conservative interactions, that is the reason why the phase image appears usually as a sort of “spatial derivative” of the topographic image, with enhanced contrast at the borders of the structures, that highlights the shape of objects that would be more difficult to tell from the sole topography. In this case, it is not correct to assign to the phase contrast any meaning related to mechanical properties, since the same contrast is observed as well on homogeneous (but rough) surfaces.

Author response: TM amplitude setpoint was automatically determined by our AFM device and was typically set in the range of 350-375 mV. We have made this clear in our methods on the sixth sentence of the first paragraph of section 2.2 of our methods. However, we dispute this point made by the reviewer. As shown previously, a rough material with homogenous material properties would not show the same contrast in the phase map as the height map, demonstrated by refs 35,36,37. The phase angle across any surface substructure with the same material properties would not vary as there would be no cause for a change in the lag or lead, as a result of any combination of repulsive or attractive forces, of the phase response.

Quantitative nanomechanical mapping (QNM) is claimed to provide more accurate information about local mechanical properties, possibly also taking into account the contribution of local morphology. Even by assuming that the outcome of this technique could provide more significant values of local nanomechanical properties, in this work QNM is applied by just sampling on a few points of the surface, and not by full mapping with nanometer-scale resolution, therefore it is difficult to correlate the nanomechanical measurement with the related morphology (that is, to compare systematically the values on top of the structures with the ones in the regions between structures).

Author response: We realize that our wording about our intentions about the use of QNM were unclear and have made corrections in our Methods throughout the third paragraph of section 2.2. Our use of QNM was only intended to provide a general guide to the relationship between phase angle and modulus. Our results indicate that greater phase angles correlate to “softer” material properties and lower phase angles correlate to “harder” material properties. However, we cannot quantify this relationship any further.

In conclusion, although a lot of interesting information on the sample morphology is provided, the main aim of the paper, that is, to gain insight on nanomechanical properties of the investigated samples, seems not supported by evidence. The assessment of nanomechanical properties cannot be demanded exclusively to TM phase imaging, but it should be supported by more thorough exploration, by QNM or other nanomechanical investigation tools, by showing the claimed correlations between structure and functionality.

Author response: We have worded our text more carefully in the fifth paragraph of the Introduction and the second, third and fourth paragraphs of the Discussion to reflect the concerns of the reviewer. It was not our exact aim to understand mechanical properties on a quantitative basis, but rather, a qualitative basis. We hope these revisions make the aims of our paper more clear. 

Round 2

Reviewer 2 Report

Most of the points raised in my previous report were successfully addressed by the Authors. The only remaining issue concerns the information content of tapping mode phase, that in my opinion is not correctly interpreted by the Authors, and this may lead to misleading conclusions, therefore some further elaboration is needed before publication. Specifically, comments to the Authors' statements in their reply are reported below. 

"Local geometry could affect the phase map of a sample if there is non-homogenous distribution of material properties along its surface."

I am afraid that this statement is not right. Phase contrast appears even on perfectly homogeneous substances, only presenting surface roughness.

"For instance, a rough surface would show variation in along its height map, but would be largely homogenous in its phase map, see refs 35,36,37."

The amount of phase contrast depends on the topographic roughness. As mentioned in my previous report, the observed contrast depends on the spatial derivative of the topography, not on the topography itself. This means that if the surface is flat, its derivative is zero, and there is no change in phase signal. If roughness is present, the phase contrast depends on the slope of the surface. By defining an aspect ratio related to roughness as the ratio between the height and the width of typical surface structures (bumps, grains, cracks etc) the related phase contrast follows such aspect ratio. If materials contrast is present, the contrast due to such materials difference "combines" to the one due to geometry.

In Ref. [35] (page 308) it is stated: "…the interaction of the
probe with the surface can dissipate energy by means
of interaction with the surface. The loss of energy,
which is dependent on the probe–surface interaction,
causes the phase of the oscillating cantilever to shift.
The energy lost by interaction of the probe and surface
can be quantified; however, attributing the energy loss
to a particular surface property has proved more difficult.
[25]."

In Ref. [25] mentioned within, the role of conservative interactions is not considered at all. From a more complete analysis, it is actually conservative interaction that produces a phase shift, since the resonance peak is shifted and therefore also the phase value corresponding to the fixed tapping mode frequency. A dissipative interaction does neither cause frequency shift nor phase shift, but only a decrease of oscillation amplitude. Since in tapping mode AFM the oscillation amplitude is kept constant by the topography adjustment feedback loop, distance is corrected to compensate such amplitude decrease, and this causes a change in conservative interaction as well. This latter change is able to cause a frequency (and therefore a phase) shift.

I regret that knowledge of some AFM-related technique is not generally adequate for a correct interpretation of results. To me, this looks true for a large number of scientific publications employing AFM-related tools for materials' characterization. Unfortunately, citation of papers that base their conclusions on incorrect or superficial interpretation of AFM-related techniques is not enough to support one's claims. General literature reviewing the basics of AFM-related techniques should be considered more carefully in order to be confident enough to support our claims. One of the reasons why misinterpretations are very common is because AFM outcomes are considered as if they were the result of a simple measurement process, while on the contrary, nanoscale tip/sample interaction and its measuring process are rather complicated and they depend critically on the nanoscale shape and structure of the probe tip, that is not under the control of the experimentalist and can vary even during the same scan due to contamination or occasional plastic impacts. This is the reason why AFM is not at all a standardized technique, and the obtained results depend heavily on data post-processing procedures. I also see responsabilities of scientific instrumentation companies in this respect.

In Ref. [36], the observed phase contrast is between a solid substrate and a liquid layer condensed on top of it. The substrate is smooth and the expected materials' contrast is extremely high (solid vs. liquid). Therefore, the role of topography is negligible in this case.

In Ref. [37], materials' contrast seems related to a generic difference in mechanical properties, and not specifically to different surface dissipation. Actually, it is well known (see e.g. Ref. [33]) that phase contrast is related to differences in surface stiffness. This is because, as mentioned above, different materials can exhibit different stiffness (i.e. conservative, or elastic, interaction) producing a phase contrast, even when dissipative interaction is the same. By the way, there is still some confusion in citation [37] (is the publication of Hutter still to be cited at this numbering?).

“While we cannot quantify the differences we observe in the phase map nor can we state whether differences in material properties are correlated to local geometry, we can say that local geometry cannot solely be the determinant of the observed phase contrast.”

This statement is not supported by evidence. I suggest to the Authors to calculate the spatial derivative of their topographic images and compare them with their phase images. If there are spots in phase images that are markedly different from the topography derivative, in that case it could be claimed, at least qualitatively, that materials contrast could be present in those locations. Please have a look, for instance, to Fig. 3 and 6 of Ref. [35]. In Fig. 3, spots in phase contrast correspond to spots in topography, however they don't look like their spatial derivative (enhanced borders), but rather, they exhibit contrast over the whole spot size. In Fig. 6 instead, contrast coincides with the derivative of topographic image (enhanced borders of structures).

"TM amplitude setpoint was automatically determined by our AFM device and was typically set in the range of 350-375 mV." 

The units indicated are arbitrary and do not provide any physical information on the amount of interaction. The relevant quantity is oscillation amplitude (in nm) compared with the free oscillation amplitude (in nm). Automatic determination of measurement parameters can be accepted as long as the criterion for such adjustment is known, that instead is not provided by the Authors. User manuals of the instrument should provide the necessary information about this issue, although this is not for granted. This is a further example of how AFM is not a standardized technique. If the Authors were to change the measurement setpoint manually, they could convince themselves on how phase results are influenced by measurement parameters in AFM.

"As shown previously, a rough material with homogenous material properties would not show the same contrast in the phase map as the height map, demonstrated by refs 35,36,37."

As shown previously, a rough material with homogeneous materials properties WOULD show the same contrast in the phase map as the DERIVATIVE of the height map. To convince themselves, the Authors could try to differentiate one of their height maps (by applying a "high pass" filter present in all AFM data processing software) and compare it to the phase image. Try to apply scale inversion if the derivative images looks like the negative of the phase image, to find the best match.

Refs. [35,36,37] by no means demonstrate the Authors' statement above. 

“The phase angle across any surface substructure with the same material properties would not vary as there would be no cause for a change in the lag or lead, as a result of any combination of repulsive or attractive forces, of the phase response.”

I regret that this statement is not correct. As discussed above, surface substructure (in my understanding, this means a geometrical structuring) with the same materials properties IS able to vary the phase angle, as explained above, by the action of conservative interactions and feedback topography adjustment.

In conclusion, I suggest that referencing to misleading literature are eliminated, incorrect statements are amended with a more rigorous consideration of the actual AFM measurement mechanisms, along the lines of proper literature, and conclusions are fixed accordingly, in order to allow publication on Molecules.
